# Spatiotemporal Differentiation and Influencing Factors of Frost Key Date in Harbin Municipality from 1961 to 2022

Tian-Tai Zhang [1,2,3], Chang-Lei Dai [1,2,3,*], Shu-Ling Li [4], Chen-Yao Zhang [1,2,3], Yi-Ding Zhang [1,2,3] and Miao Yu [1,3,5,6,*]

1   School of Hydraulic and Electric-Power, Heilongjiang University, Harbin 150080, China; hss_zhangtiantai@126.com (T.-T.Z.); hss_zhangchenyao@163.com (C.-Y.Z.); 13936500425@163.com (Y.-D.Z.)
2   Institute of Groundwater in Cold Regions, Heilongjiang University, Harbin 150080, China
3   Heilongjiang Joint Laboratory of Hydrology and Hydraulic Engineering in Cold Region, Harbin 150080, China
4   Harbin Meteorological Bureau, Harbin 150080, China; lishuling1968@163.com
5   Faculty of Geology and Survey, M. K. Ammosov North-Eastern Federal University, Yakutsk 677000, Russia
6   Melnikov Permafrost Institute of the Siberian Branch of the Russian Academy of Science, Yakutsk 677000, Russia
*   Correspondence: daichanglei@126.com (C.-L.D.); hss_yumiao@126.com (M.Y.)

**Abstract:** This study analyzed frost formation data provided by the Harbin Meteorological Bureau and considered geographic factors, temperature, and population density. Various analytical methods, including linear fitting, the Mann–Kendall mutation test, the Pettitt method, and the sliding *t*-test, were employed to identify the temporal and spatial changes as well as the effects of these factors on frost dates in Harbin. The study shows that the first FSD occurred on 18 August, in both 1966 and 1967, which was the 255th day. The latest FSD was observed on 10 October 2006, which was the 283rd day. The earliest occurrence of an FED was on 24 April 2015, which was the 114th day, and the latest was on 21 April 1974, which was the 141st day. The highest number of frost days occurred in 2012, with 161 days, whereas the shortest year was 1966, with only 123 frost days. Throughout the study period, the FSD increased by 7.8 days at a rate of $-1.27\text{d}/10\text{a}$, the FED increased by 10.9 days at a rate of $1.77\text{d}/10\text{a}$, and the FFS increased by 18.9 days at a rate of $3.05\text{d}/10\text{a}$. The propensity rates of the FSD and FFS at each location in Harbin indicate an upward trend, while for the FED, certain locations display an upward trend. In general, the FSD has exhibited a delayed trend, the FED has shown an earlier trend, and the FFS has experienced an extended trend. With one-way linear regression, the FSD exhibited an increasing trend at each site, while the FFS also indicated a similar trend, and the FED showed an overall decreasing trend. Throughout the study period, a change was observed in the FSD in 2000, resulting in an average arrival time of the 265th day, or 22 September, of that year. Subsequently, post mutation, the average arrival time of the FSD in the study area was the 272nd day, or 29 September, of that year. In 2006, the FED also underwent a change, with the average arrival time in the study area being the 128th day, or 4 April, of that year. After the change, the average arrival time of the FED in the study area was the 121st day, i.e., 8 April. In 1 April 2004 of that year saw a change in the FFS. Prior to the change, the FFS in the study area averaged the 137th day, whilst following the change, the FFS in the study area averaged the 150th day. The FSD and FFS within Harbin exhibit a negative correlation with latitude and a positive correlation with temperature. Additionally, the FED displays a positive correlation with latitude and a negative correlation with temperature. As the FSD, FED, and FFS in central Harbin are the earliest, latest, and longest, the Pearson correlation coefficient method and multiple regression cannot adequately reflect the effect of longitude.

**Keywords:** frost; cold region; M-K test; meteorological factors; temporal and spatial variations

## 1. Introduction

Harbin is situated in the frigid high-latitude zone [1]. In recent years, frost has been a significant agricultural hazard in the study area. Frost variations are mainly influenced by large-scale atmospheric circulation and atmospheric oscillations [2–4]. However, there is a lack of sufficient studies characterizing the variability of critical frost dates in relation to agricultural practices and spatial factors across China, particularly in the northeast. Therefore, this study utilizes meteorological data from 1961 to 2022. Linear fitting and Mann–Kendall mutation tests are applied to ascertain the changing characteristics of frost occurrence in terms of date and time in Harbin. In addition, this study analyzed the geographic factors, temperature, and population density using Pearson's method and multiple linear regression. With reference to previous studies, this study adopts the minimum surface temperature of $\leq 0\,^{\circ}\text{C}$ as the frost index [5].

In recent years, the global temperature has seen a rise, with the temperature in the 2010s registering an increase of $1.09\,^{\circ}\text{C}$ compared with the mid-to-late 19th century [6]. The spatiotemporal variation of frost and the ensuing extreme weather-related disasters have attracted wide attention from scholars around the world. The number of annual frost days has decreased significantly across most parts of mainland China, as has occurred in most other parts of world [7,8]. To date, research on frost has been limited in comparison to that conducted on snow and permafrost. In recent years, researchers have focused on the variations of near-surface soil freezing conditions. For instance, the timing and duration of such freezing phenomena in China have been studied [9], as well as the annual/monthly area range of seasonal soil freezing conditions in China [10]. These investigations primarily deduced the state and occurrence of soil freezing through soil temperature fluctuations and air or surface temperatures. These factors also affect the development of frost damage. Snow is of key importance to the water and energy cycle [11]. In cold regions, both the snow depth and the timings of snow build-up and melting have considerable impacts on ecosystem dynamics and biological activities including plant growth [12], carbon balance [13], animal activity [14], biodiversity patterns [15], etc. As a result of recent global warming, snow melt has begun to advance, leading to a significant increase in springtime carbon sequestration across the majority of boreal regions in the Northern Hemisphere [16]. Snow and ice are essential components of the global hydrological and energy cycles, and they are closely associated with frost occurrence [17]. Frost has an important impact on the regulation of a municipality's reservoir and constitutes a vital component of hydrological facilities, infrastructure construction, and winter tourism, among others [18]. The FSD, FED, and growing season length exert a notable influence on annual runoff, drought frequency, evaporation, and other factors [19]. Extreme disasters can lead to differential nutrient accessibility for the same plants and causes differences in plant growth [20] and reproduction [21]. Frost-induced damage can give rise to huge harms to wheat [22–24], persimmon [25], broad beans [26], and other crops [27–30]. The long-lasting exposure of plants to frost during the growing period greatly affects not only economic loss but also the ecological environment. In Europe, early spring warming results in a "false spring", rendering plants more vulnerable to frost [31]. Some scholars argue that early spring can lead to premature development of plant organs and expose vulnerable plant organs to cold weather, ultimately causing greater damage [32]. In the United States, late frost can have varying effects on different species of Ozark trees. However, damage estimation is difficult due to the lack of a significant link between damage and subsequent growth [33].

At the same time, frost also affects the impact resistance of reinforced concrete beams [34]. This results in the carbonation of reinforced concrete to reduce the durability of the original structure [35]. Frost damage affects the viscosity of coarse-aggregate reinforced concrete and ordinary reinforced concrete [36,37]. Additionally, pervious concrete is more susceptible to damage in frost conditions [38].

With the global warming trend, all regions are facing an increasing FFS. In 1973, the annual number of frost days in China was about two days higher than the average between

1961 and 1990. From 1973 to 1985, the annual number of frost days remained close to the average level, with minimal inter-annual variation. However, after 1985, the annual number of frost days decreased rapidly, and a clear reversal was observed around 1992. The initial extension of FFS lagged behind the rapid increase in daily minimum temperatures by about ten years, while the decrease in the number of annual frost days lagged by about 15 years [39]. Altitude accounts for 96–100% of the variation in frost day occurrences. The vertical gradient of a frost day is about seven days/100 m. Between 1936 and 2013, the long-term trend of frost days was not significant, and the rate of change in frost days was basically negative, not less than −1.5 days per decade. During the most intense global warming, the rate of change in frost days is less than −3 days per decade and, in some cases, equal to −15 days per decade [40]. Due to the unique topography of the Libyan Peninsula (IP), frost days exhibit wide variability, with the general trend being 0.4–1.6 days later for FSD per day and 0.42–1.29 days earlier for FED per year [41]. In the United States from 1948 to 1999, FSD showed a tendency for delay, FED showed a trend of advance, and FFS exhibited a trend of extension [42]. Despite the increasing temperatures in South Korea over the years, frost day and FSD have neither increased nor been delayed, and even the last frost date has been delayed by 0.5 days per year [43]. In Japan, the first frost date is delayed by 0.224 days per year, and the last frost date is advanced by 0.228 days per year. Sites with strong winds contributed to a decrease in the temperature criterion [44].

So far, few efforts have been made to describe the nature of these changes in space and time, especially in relation to other variables. This study analyses and discusses geographical factors, temperature, and population density. Understanding the changes in FSD, FED, and FFS in Harbin is of great significance for research on global warming and its impact on agricultural production. The aim of this study is twofold: (1) to analyze the spatiotemporal variations of FSD, FED, and FFS indices in Harbin between 1961 and 2022, and (2) to investigate the correlation between FSD, FED, and FFS and temperature and population density in Harbin.

## 2. Study Area, Data Sources, and Methods

### 2.1. Study Area

The research site is situated in Harbin, Heilongjiang Province, China (125°42′–130°10′ E and 44°04′–46°40′ N) and is considered a typical municipality in the high-latitude cold zone. It covers an area of roughly 53,000 square kilometers, with an average elevation of 118.3 m. The climate of the area is characterized by a mid-temperate continental monsoon, with an average annual temperature of approximately 4.1 °C. Moreover, there is an average annual maximum temperature of 9.6 °C, an average annual minimum temperature of −2.2 °C, an average annual wind speed of roughly 5.5 m/s, and an average annual humidity of around 70% [1,45]. Figure 1 illustrates the geographical location of the research area, including the geomorphological features and the positions of meteorological stations. The codes and names of each meteorological station are provided in Table 1.

**Table 1.** Names of different meteorological stations.

| Station | Name | Station | Name |
|---|---|---|---|
| 50867 | Bayan | 50962 | Mulan |
| 50877 | Yilan | 50963 | Tonghe |
| 50953 | Harbin | 50964 | Fangzheng |
| 50955 | Shuangcheng | 50965 | Yanshou |
| 50956 | Hulan | 50968 | Shangzhi |
| 50958 | Acheng | 54080 | Wuchang |
| 50960 | Binxian | —— | —— |

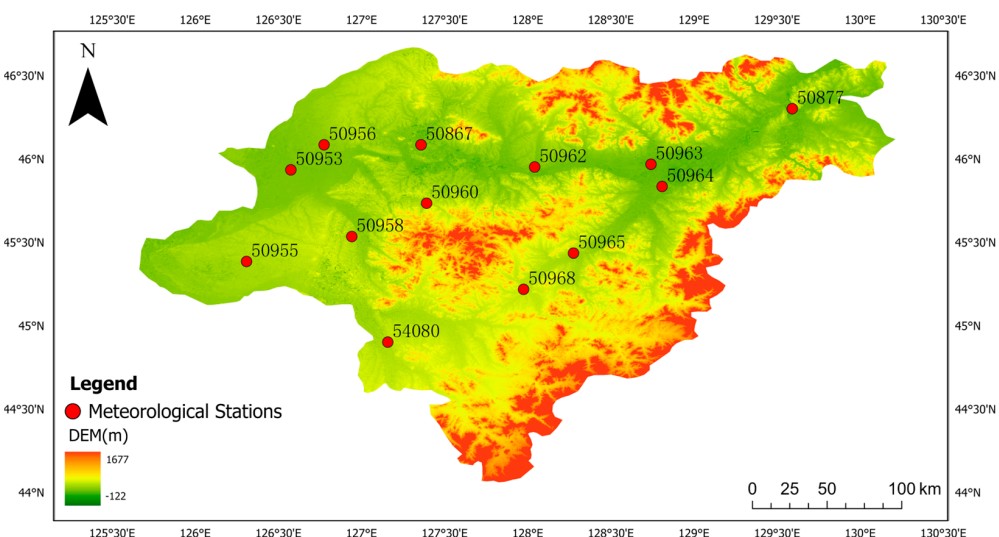

**Figure 1.** Study area and distribution of meteorological stations.

### 2.2. Data Sources

It has been found that in northern China, the formation of frost typically occurs after August 1. Therefore, the first occurrence of frost formation after August 1 is defined as the frost start date (FSD), while the final occurrence of frost before August 1 is denoted as the frost end date (FED). The period between the FED and FSD is referred to as the frost-free season (FFS). This article presents a method for converting each year's date into a numerical symbol. For instance, January 1st is represented as 1, March 1st as 60, and in leap years, March 1st is represented as 61. The method avoids subjective evaluations and uses clear, objective language to maintain a formal register.

The meteorological data utilized in this research comprises the 0 cm surface air temperatures from 13 meteorological stations in Harbin, obtained from the Harbin Meteorological Bureau. The study employs a 30 m spatial resolution digital ground model offered by the Japan Aerospace Exploration Agency (JAXA) to collect digital elevation data (DEM) for Harbin. Population density data were gathered from seven censuses in China. The data from the first to sixth censuses were sourced from the Scientific Data of China website [46] (http://www.csdata.org/p/574/ (accessed on 5 August 2023)), and the data from the seventh census were obtained from the official website of the Harbin Municipal People's Government (https://www.harbin.gov.cn/haerbin/c104569/202105/c01_70396.shtml (accessed on 5 August 2023)).

### 2.3. Methods

In this study, the trends of key variables were analyzed by linear trend fitting, their significance was assessed through the Mann–Kendall mutation test, and the authenticity of mutation points was verified by the Pettitt method and sliding *t*-test. The ordinary kriging interpolation method is used to interpolate the key frost variables from a point scale to a regional scale. Considering that the factors affecting the key variables of frost include not only latitude and longitude but also altitude, this study takes altitude as a covariate to improve the longitude of interpolation. Moreover, the Pearson correlation coefficient method and multiple linear regression were utilized to calculate the correlation and regression of elevation, longitude, latitude, air temperature, and population density with the key frost variables. In this paper, FSD, FED, and FFS are spatially analyzed using Kriging linear interpolation.

## 3. Results

### 3.1. Temporal and Spatial Variation and Mutation Test of FSD

Figure 2 displays the spatial distribution of the FSD in Harbin from 1961 to 2022. As shown in Figure 3, the FSD arrives earlier in the south-central region of Harbin and later in the western section, with a general trend towards gradual advancement from the east and west ends towards the center. The FSD in Harbin typically occurs between the 264th and 272nd day of the year, corresponding to 21 September to 29 September. Figure 4 illustrates the overall variation in FSD in Harbin, which exhibits a consistent trend of delay over the years. The FSD increased by 7.8 days at $-1.27$ d/10a throughout the study period. The earliest FSD occurred on the 255th day of 1966, which was 18 September, and the latest on the 283rd day of 2006, which was 10 October. The Mann–Kendall mutation test was employed to examine the significance of the FSD change trend. The Pettitt method and the sliding $t$-test were used to verify the authenticity of the mutation points. The Mann–Kendall mutation test revealed that FSD underwent changes in 2000. Both the Pettitt method and sliding $t$-test confirmed the statistical significance of the mutation points at $\alpha = 0.05$, indicating noticeable differences at the mutation points. Prior to the changes, the FSD's average arrival time in the study area was 265 days, which is 22 September of the year. Following the changes, the average arrival time increased to the 272nd day, i.e., 29 September of that year.

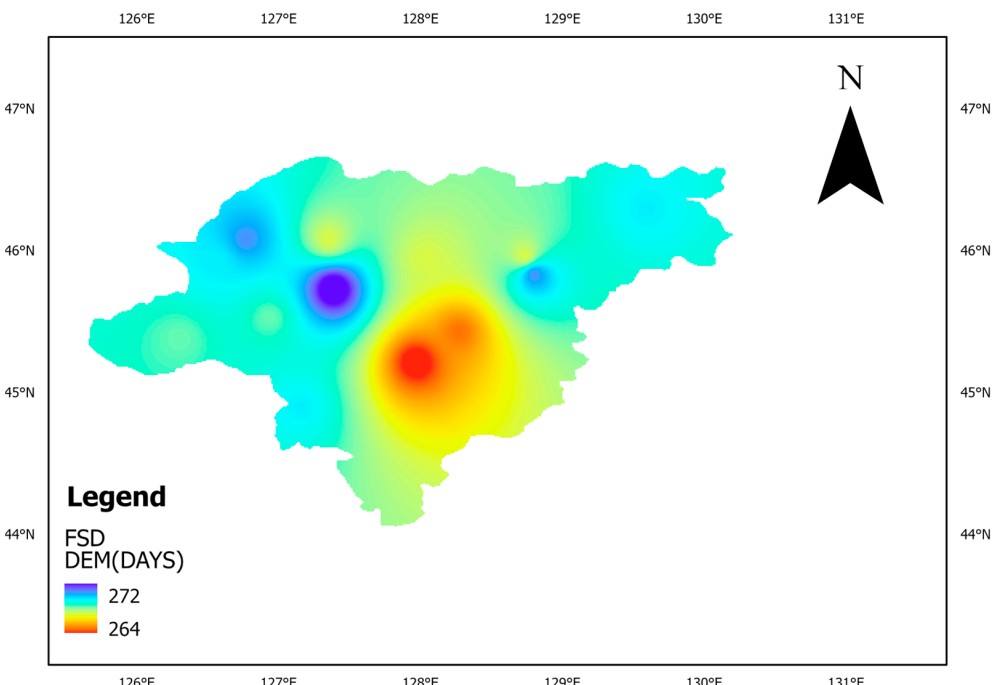

**Figure 2.** Spatial distribution of FSD in Harbin Municipality from 1961 to 2022.

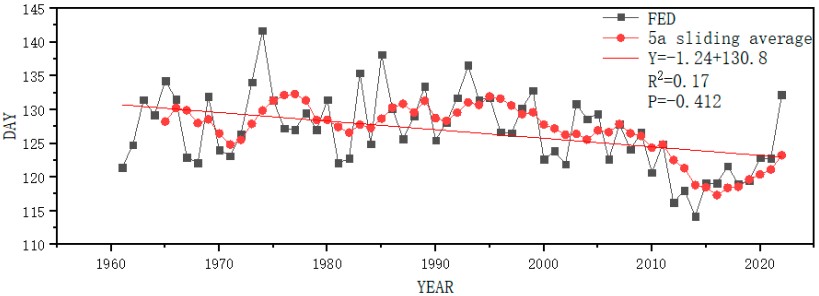

**Figure 3.** Interannual variation in FSD in Harbin Municipality.

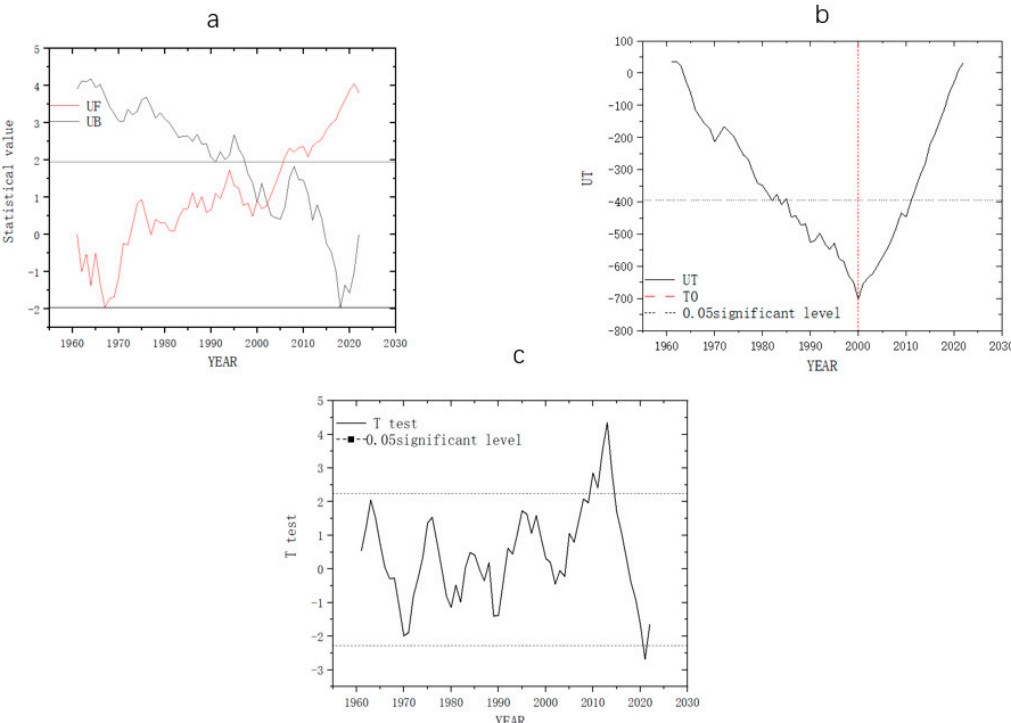

**Figure 4.** M-K (**a**), Pettitt (**b**), and sliding *t*-test (**c**) of FSD in Harbin Municipality.

### 3.2. Temporal and Spatial Change and Mutation Test of FED

Figure 5 shows the spatial distribution of the FED in Harbin during 1961–2022. As shown in Figure 6, the FED arrived later in the south-central part of Harbin and earlier in the west. The overall trend indicates a gradual postponement from the east and west ends toward the center. The FEDs in Harbin occur between the 121st and 135th days, i.e., from 1 April to 15 April. Figure 7 illustrates the overall change in the FED in Harbin, with a trend of delay from year to year. The FED increased by 10.9 days at a rate of 1.77d/10a throughout the study period. The earliest FED occurred in 2014, on the 114th day of the year, i.e., 24 April of that year, while the latest was in 1974, on the 141st day of the year, i.e., 11 May of that year. The Mann–Kendall mutation test was conducted to check the significance of the change trend of the FED, and we tested the integrity of the mutation points using the Pettitt method and the sliding *t*-test. The Mann–Kendall mutation test confirmed that the change in FED in 2000 was statistically significant, and the mutation point passed the significance test of $\alpha = 0.05$ for both the Pettitt method and sliding *t*-test, indicating significant differences at the mutation points. Before the change, the average arrival time of the FED in the study area was the 128th day, i.e., 8 April. After the change, the arrival time of FED averaged 121st d, i.e., 1 April of the year.

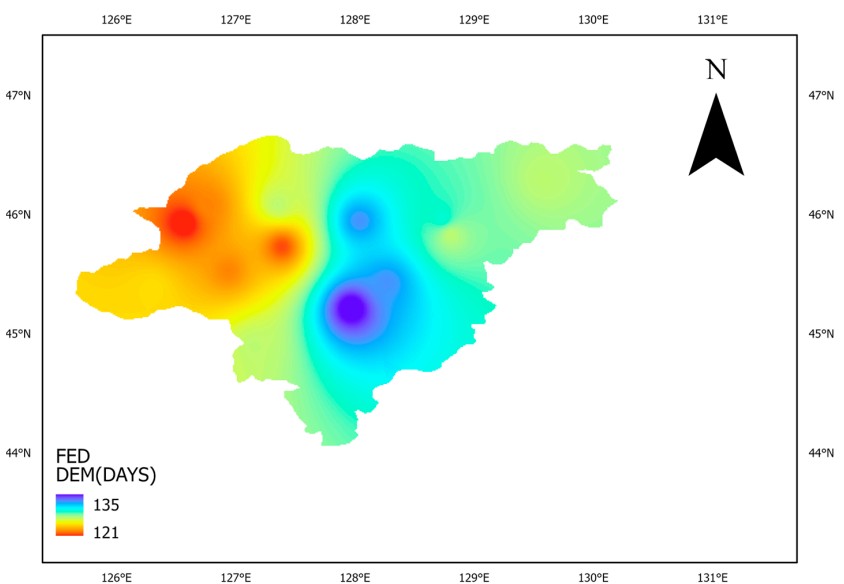

**Figure 5.** Spatial distribution of FED in Harbin Municipality from 1961 to 2022.

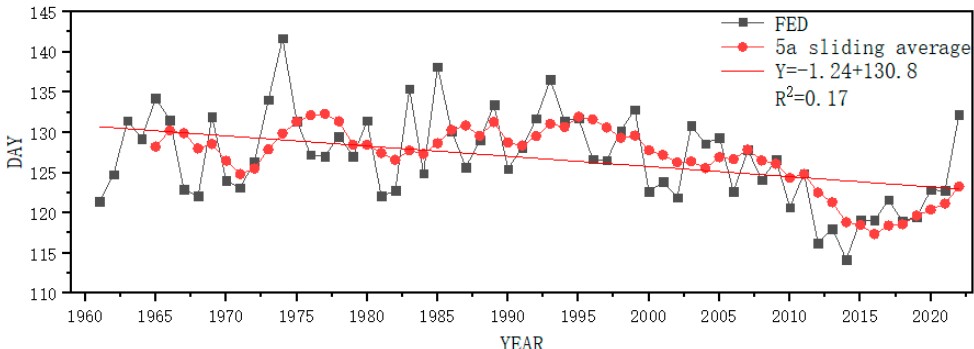

**Figure 6.** Interannual variation in FED in Harbin Municipality.

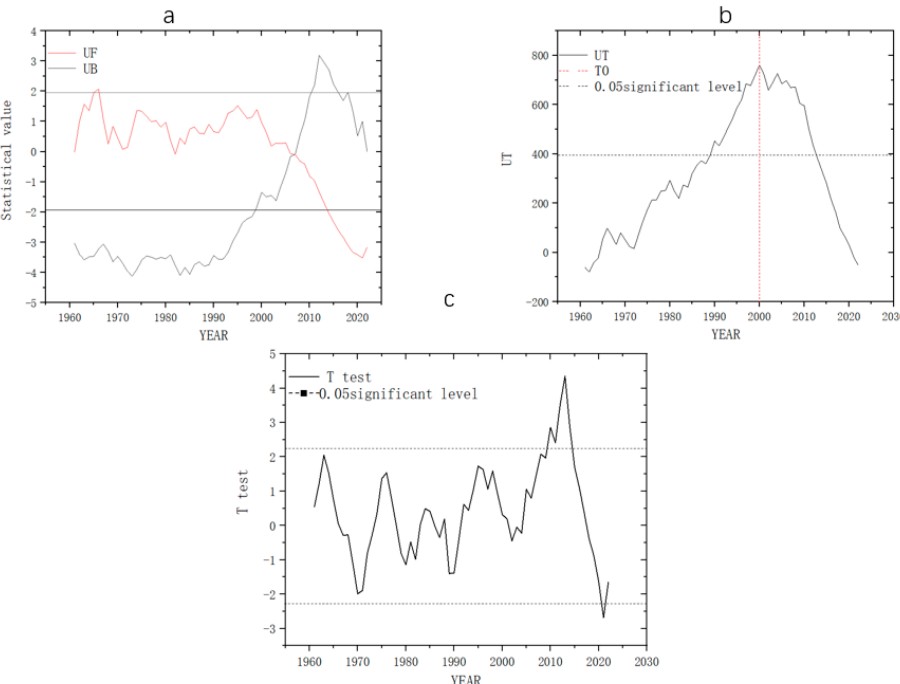

**Figure 7.** M-K (**a**), Pettitt (**b**), and sliding *t*-test (**c**) of FSD in Harbin Municipality.

### 3.3. Test of Spatio-Temporal Variation and Abrupt Change in FFS

Figure 8 illustrates the geographical distribution of the FFS in Harbin between 1961 and 2022. Figure 9 indicates that the FFS is shorter in the south-central region of Harbin, whilst longer in the western part. The general trend is a decrease from the east and west towards the center, and the number of days of the FFS falls within the range of 129–149 days. Figure 10 displays the overall change in the FFS in Harbin, depicting a steady increase in length over the years. The FFS increased by 18.9 days over the study period, at a rate of 3.05d/10a. The highest FFS was recorded in 2012 at 161 days, and the shortest in 1966 at 123 days. To verify the significance of the FFS trend and identify mutation points, we conducted the Mann–Kendall mutation test, Pettitt method, and sliding *t*-test. The Mann–Kendall mutation test indicated that a change in the FFS occurred in 2004. In addition, the Pettitt method and sliding *t*-test showed significant differences among mutation points at $\alpha = 0.05$. The study area had an average FFS of 137 days before the change but saw an increase to an average of 150 days after the change.

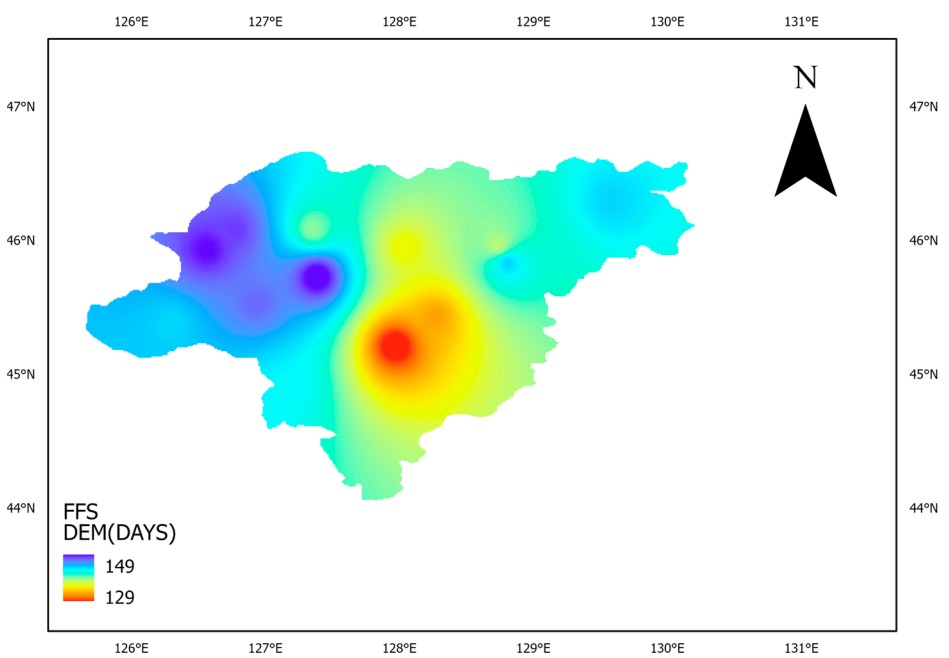

**Figure 8.** Spatial distribution of FFS in Harbin Municipality from 1961 to 2022.

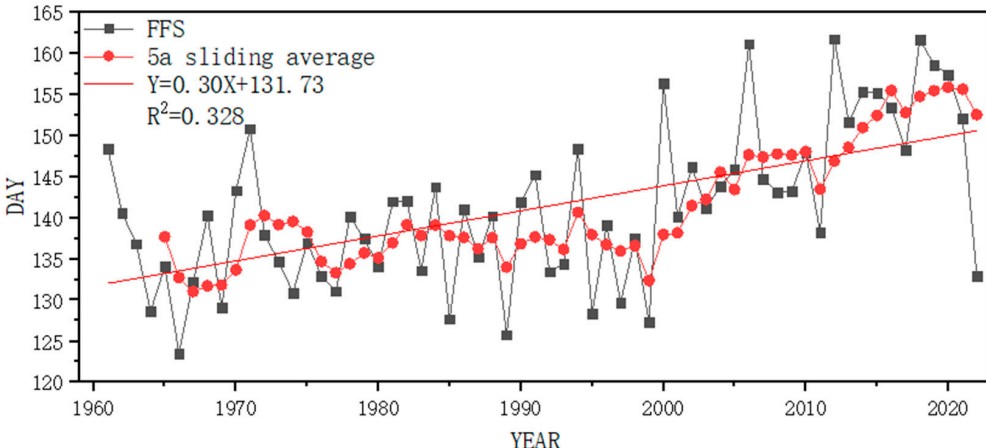

**Figure 9.** Interannual variation in FED in Harbin Municipality.

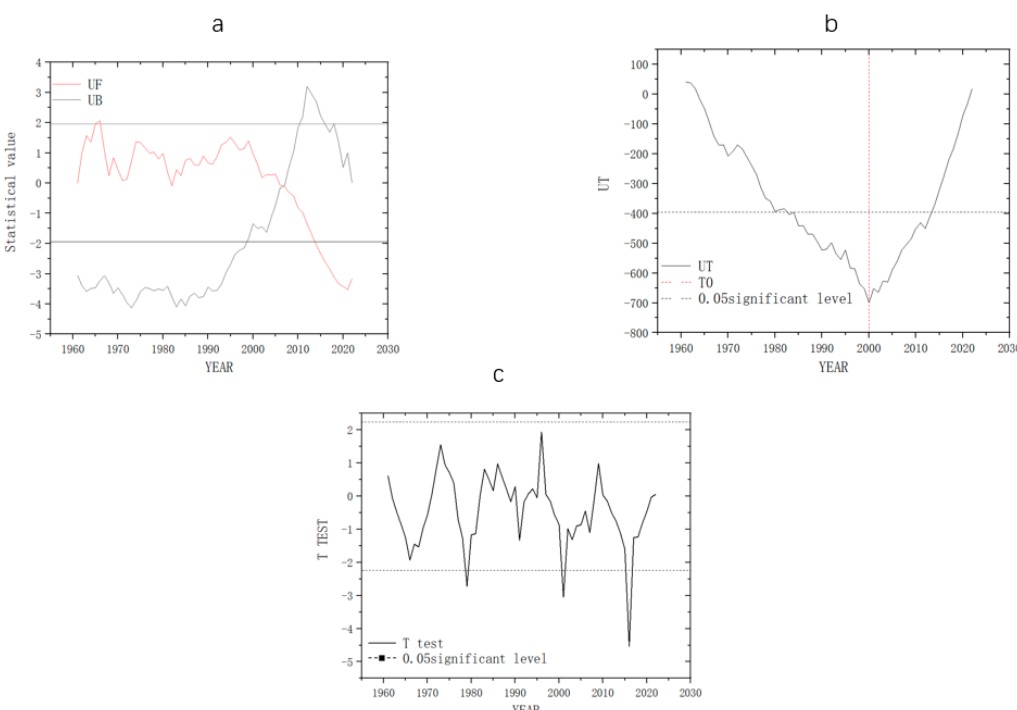

**Figure 10.** M-K (**a**), Pettitt (**b**), and sliding *t*-test (**c**) of FFS in Harbin Municipality.

### 3.4. Factors Influencing of Frost

The correlation coefficients for FSD, FED, and FFS in relation to geographical factors, mean temperature, and population density in Harbin are presented in Table 2. The correlation coefficient between FSD and temperature r is 0.363, whereas for FED and FFS, the coefficients are −0.614 and 0.557, respectively. The correlations between FED and FFS and temperature are statistically significant, whereas their correlations with both geographical factors and population density are deemed insignificant. The regression coefficients indicate that a 1° increase in latitude corresponds to an advancement of the FSD by 1.44 days, a delay of the FED by 5.15 days, and a prolongation of the FFS by 6.6 days. Additionally, a 1 °C rise in mean temperature led to an FSD advancement of 1.38 days, a delay of the FED by 3.026 days, and a prolongation of the FFS by 4.409 days. In Harbin, latitude exhibits a negative correlation with the FSD and FFS while showing a positive correlation with temperature. On the other hand, the FED demonstrates a positive correlation with latitude and a negative correlation with temperature. In the central part of Harbin Municipality, FSD, FED, and FFS occur at their earliest, latest, and longest points, respectively. Therefore, the influence of longitude cannot be captured using the Pearson correlation coefficient and multiple regression analysis methods.

**Table 2.** Relationships of the FSD, FED, and FFS with the geographical parameters, average temperature, and population in Harbin Municipality.

| Factor | IF | FSD | FED | FFS |
|--------|------|--------|---------|--------|
|        | ASL  | −0.05  | −0.048  | 0.016  |
|        | LAT  | 0.523  | −0.121  | 0.165  |
| CC     | LON  | 0.147  | −0.134  | 0.145  |
|        | TEMP | 0.363  | **−0.614** | **0.557** |
|        | POP  | 0.187  | −0.514  | 0.426  |
|        | ASL  | 0.007  | −0.028  | 0.036  |
|        | LAT  | 1.314  | −0.4164 | 5.478  |
| RG     | LON  | −0.013 | 1.505   | −1.519 |
|        | TEMP | 1.383  | −3.026  | 4.409  |
|        | POP  | ~0     | −0.002  | 0.002  |

## 4. Discussion

In this analysis, we have amassed 62 years' worth of frost day data in Harbin from 1961 to 2022. The majority of annual frost-free seasons (FFSs) span roughly 140 days from mid–late May to mid–early September, whilst the frost start date (FSD) occurs in mid-to-late August and the frost end date (FED) in mid-to-early May. From the late 1950s to the 1960s, the FSD progressed from late August to early September, and the FED was delayed until early June each year. The FSD was first recorded in 1966 and 1967, both on the 255th day, which is 18 August; the most recent occurrence was in 2006 on the 283rd day, which is 10 October. FED was first recorded on the 114th day, which is 24 April, in 2015, and most recently in 1974, on the 141st day, 21 April of that year. The year with the highest number of frost days was 2012, with 161 days, and the shortest year was 1966, with 123 days.

### 4.1. Trends in FSD, FED, and FFS

As shown in Table 3, the propensity rates for the FSD and FFS exhibited an increasing trend for each site in Harbin. For the FED, certain sites also demonstrated an increasing trend in the propensity rate. In general, the FSD displayed a tendency towards delay, while the FED showed a propensity towards advancement, and the FFS exhibited a tendency towards extension. Analysis using the method of unary linear regression revealed an increasing tendency for both the FSD and FFS at all sites, coupled with an overall decreasing tendency for the FED.

**Table 3.** Variations in the FSD, FED, and FFS in different meteorological stations of Harbin Municipality from 1961 to 2022.

| Meteorological Stations | FSD | | FED | | FFS | |
|---|---|---|---|---|---|---|
| | Ave. (d) | Tre. (d/10a) | Ave. (d) | Tre. (d/10a) | Ave. (d) | Tre. (d/10a) |
| 50867 | 267 | 2.14 | 127 | −1.87 | 139 | 4.01 |
| 50877 | 269 | 2.22 | 127 | −1.64 | 143 | 3.86 |
| 50953 | 269 | 2.01 | 121 | −2.23 | 148 | 4.24 |
| 50955 | 268 | 1.64 | 125 | 0.8 | 143 | 0.84 |
| 50956 | 270 | 1.52 | 123 | 0.27 | 147 | 1.25 |
| 50958 | 268 | 1.06 | 122 | −0.21 | 146 | 1.27 |
| 50960 | 272 | 0.61 | 122 | 0.18 | 149 | 0.43 |
| 50962 | 267 | 2.09 | 132 | −2.22 | 136 | 4.31 |
| 50963 | 267 | 1.92 | 129 | −2.72 | 138 | 4.63 |
| 50964 | 270 | 1.59 | 127 | −1.48 | 143 | 3.07 |
| 50965 | 265 | 1.78 | 132 | −1.93 | 133 | 3.71 |
| 50968 | 264 | 2.91 | 135 | −2.74 | 129 | 5.65 |
| 54080 | 269 | 1.59 | 127 | −0.73 | 142 | 2.32 |
| Ave. | 268 | 1.77 | 127 | −1.27 | 141 | 3.05 |

Using the mean values of the frost elements at each station, it can be seen that the FSD occurred the earliest at weather station 50968 on the 264th day, i.e., 21 September, and the latest at weather station 50956 on the 270th day, i.e., 27 September of the mean year. The FED appeared earliest at 50953 on the 121st, i.e., May 1st of the mean year, and the latest at 50968, on the 135th, i.e., 15 March of the mean year. The highest number of FFS occurrences was at weather station 50960, totaling 149 days, whereas the lowest was at 50948, with only 129 days.

Amidst global warming, Harbin undergoes a delayed FSD, advanced FED, and extended FFS, which will extend the growing season of crops, provide sufficient heat resources for regional agricultural production, and facilitate the expansion of suitable planting areas. Heilongjiang Province is a major grain-producing area in China, and the sensitivity of crop growth to frost varies with different crops and climatic conditions. Therefore, it is necessary to implement different systems and methods to reduce potential frost damage to crops.

*4.2. Testing FSD, FED, and FFS Based on Grey Relational Analysis*

Therefore, this study adopts the grey correlation analysis method to assess several factors. The resulting grey correlation coefficients are shown in Table 4. It is observed that longitude exhibits a high grey correlation with FSD, FED, and FFS. As longitude does not display a linear relationship with FSD, FED, and FFS, longitude is not considered significant when using the Pearson method and the multiple regression method.

**Table 4.** Grey correlation coefficient of the FSD, FED, and FFS with the geographical parameters, average temperature, and population in Harbin Municipality.

| Factor | IF | FSD | FED | FFS |
|:---:|:---:|:---:|:---:|:---:|
| | ASL | 0.942967 | 0.939464 | 0.951001 |
| | LAT | 0.987238 | 0.975974 | 0.992901 |
| GCC | LON | 0.988881 | 0.97644 | 0.993903 |
| | TEMP | 0.958773 | 0.971249 | 0.960265 |
| | POP | 0.848286 | 0.847311 | 0.857801 |

*4.3. Study Limitations*

This research is specific to Harbin, which presents an unusual observation. In the central part of the city, the first frost-free day (FSD) arrives earlier, the final frost-free day (FED) arrives later, and the frost-free season (FFS) lasts longer compared to findings in many previous studies. This discrepancy can be attributed to the relatively small study area with certain peculiarities and limitations. Secondly, this unique occurrence is influenced by the smaller population in the central part of Harbin, which explains the observed phenomenon.

**5. Conclusions**

Through examining the spatial and temporal fluctuations of the FSD, FED, and FFS in Harbin and their associations with geographic factors, temperature, and population density, the findings indicate that:

(1) The first FSD occurred on 18 August, in both 1966 and 1967, which was the 255th day. The latest FSD was observed on 10 October 2006, which was the 283rd day. The earliest occurrence of an FED was on 24 April 2015, which was the 114th day, and the latest was on 21 April 1974, which was the 141st day. The highest number of frost days occurred in 2012, with 161 days, whereas the shortest year was 1966, with only 123 frost days.

(2) Throughout the study period, the FSD increased by 7.8 days at a rate of $-1.27d/10a$, the FED by 10.9 days at a rate of $1.77d/10a$, and the FFS by 18.9 days at a rate of $3.05d/10a$. The propensity rates of the FSD and FFS at each location in Harbin indicate an upward trend, while for the FED, certain locations display an upward trend. In general, the FSD has exhibited a delayed trend, the FED has shown an earlier trend, and the FFS has experienced an extended trend. With one-way linear regression, the sites exhibit an upward trend for both the FSD and FFS, and a downward trend for the FED altogether.

(3) Throughout the study period, a change was observed in the FSD in 2000, resulting in an average arrival time of the 265th day, or 22 September, of that year. Subsequently, post mutation, the average arrival time of the FSD in the study area was the 272nd day, or 29 September, of that year. In 2006, the FED also underwent a change, with the average arrival time in the study area being the 128th day, or 4 April, of that year. After the change, the average arrival time of the FED in the study area was the 121st day, i.e., 8 April. April 1st of that year saw a change in the FFS in 2004. Prior to the change, the FFS in the study area averaged 137 days, whilst following the change, the FFS in the study area averaged 150 days.

(4) The FSD and FFS within the Harbin area exhibit a negative correlation with latitude and a positive correlation with temperature. Additionally, the FED displays a positive

correlation with latitude and a negative correlation with temperature. As the FSD, FED, and FFS in central Harbin are the earliest, latest, and longest, the Pearson correlation coefficient method and multiple regression cannot adequately capture the effect of longitude. Several factors were evaluated using the grey correlation analysis method. The longitudinal variables demonstrate a significant relationship with the FSD, FED, and FFS.

**Author Contributions:** Conceptualization, T.-T.Z. and C.-L.D.; software, T.-T.Z.; validation, C.-L.D., C.-Y.Z. and Y.-D.Z.; formal analysis, M.Y.; investigation, C.-Y.Z. and Y.-D.Z.; resources, C.-L.D.; data curation, S.-L.L.; writing—original draft preparation, T.-T.Z.; writing—review and editing, M.Y. and C.-Y.Z.; visualization, T.-T.Z. and C.-Y.Z.; funding acquisition, C.-L.D. All authors have read and agreed to the published version of the manuscript.

**Funding:** This study was supported by the 2020 Higher Education Teaching Reform Key Commissioned Project, (Grant No. SJGZ20200135) and the Innovative Research Project for Graduate Students of Heilongjiang University (Grant No. YJSCX2021-083HLJU).

**Data Availability Statement:** Frost key data have been uploaded. Population density data were obtained from seven censuses in China, of which the first through sixth census datasets were from the website Scientific Data of China (http://www.csdata.org/p/574/ (accessed on 5 August 2023)), and the data of the seventh census were from the official website of the Harbin Municipal People's Government (https://www.harbin.gov.cn/haerbin/c104569/202105/c01_70396.shtml (accessed on 5 August 2023)).

**Conflicts of Interest:** The authors declare no conflict of interest.

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
