# Peer review of "Spatiotemporal Differentiation and Influencing Factors of Frost Key Date in Harbin Municipality from 1961 to 2022"

_water, doi:10.3390/w15193513_

Round 1
Reviewer 1 Report
Identifying how the freeze-thaw cycle responses to climate change in important to understanding the impact of warming in cold regions. In this study, Zhang et al use observations to investigate the frost date changes in Harbin Municipality and its possible drivers. This study fits the scope of the journal. The manuscript is nicely written. I think it is acceptable after “Minor Revision”.
Comments:
1) Throughout the manuscript, please add the statistical significance test (e.g., p values) for the trends shown in this study. In addition to test, this comment also includes the figure 2, 6, 9.
2) I understand Figure 2, 6, 9 well. But this sort of figures are all for regional-mean results. Can you please show the spatial pattern for the trends of FSD, FED, FFS? Just like Figure 5 but for trends.
3) More references about the phenology change of cryosphere elements (e.g., frozen soil, snow and others) should be reviewed in the Introduction part. This includes, but not limited to, two typical studies: https://doi.org/10.1016/j.gloplacha.2018.09.016 and https://doi.org/10.1007/s00382-020-05422-z These kinds of studies could provide more background of the manuscription.
4) For title: I cannot well understand “frost key date”. How about frost phenology? Also please change “recent 62a” to the specific period you investigated. This is more clear for readers.
5) For data: any there any data gaps? How about the quality control of the data? Do they have well homogeneity?
Author Response
Dear Editors and Reviewers:
Thank you for your letter and for the reviewers’ comments concerning our manuscript entitled"Spatiotemporal differentiation and influencing factors of frost key
date in Harbin municipality in recent 62a" (water-2578506).Those comments are all valuable and very helpful for revising and improving our paper. We have studied comments carefully and have made correction which we hope meet with approval. The main corrections in the paper and the responds to the reviewer's comments are as flowing:
Reviewer 1
â‘ The review's comment: Throughout the manuscript, please add the statistical significance test (e.g., p values) for the trends shown in this study. In addition to test, this comment also includes the figure 2, 6, 9;
The authors' answer: For Figures 3, 6 and 9, P-values have been added using statistical methods and the P-values are given in the legend of the icons.
â‘¡The review's comment: I understand Figure 2, 6, 9 well. But this sort of figures are all for regional-mean results. Can you please show the spatial pattern for the trends of FSD, FED, FFS? Just like Figure 5 but for trends.
The authors' answer: The charts displaying spatial trends have recently been uploaded;
â‘¢The review's comment: More references about the phenology change of cryosphere elements (e.g., frozen soil, snow and others) should be reviewed in the Introduction part. This includes, but not limited to, two typical studies: https://doi.org/10.1016/j.gloplacha.2018.09.016 and https://doi.org/10.1007/s00382-020-05422-z These kinds of studies could provide more background of the manuscription. The authors' answer: Some additional research has been incorporated into the introduction, as indicated in lines 40 to 55, approximately;
â‘£The review's comment: For title: I cannot well understand “frost key date”. How about frost phenology? Also please change “recent 62a” to the specific period you investigated. This is more clear for readers.
The authors' answer: Revise the thesis title to Spatiotemporal differentiation and influencing factors of frost key date in Harbin municipality from 1961 to 2022
⑤The review's comment: For data: any there any data gaps? How about the quality control of the data? Do they have well homogeneity?
The authors' answer: The data is real and valid and I will submit it to the database.
We tried our best to improve the manuscript and made some changes in the manuscript.
We appreciate for Editors/Reviewers’warmwork earnestly, and hope that the correction will meet with approval. Once again thank you very much for your comments and suggestions.
Yours sincerely,
Tiantai Zhang
Reviewer 2 Report
I recommend the authors consider the following suggestions:
1. In the introduction section, the progress on frost study in the world should be summarized and highlighted the key finding. The methodology for frost study also should be introduced. Why you conducted this study was also presented clearly.
2. In the study area data sources and methods section, method should be presented clearly.
3. In the discussion section, some limitation about this study should be discussed.
I think there are a big room to improve the language in your article. Your article may be polished by a native English-speaking person and then submitted to the Journal.
Author Response
Dear Editors and Reviewers:
Thank you for your letter and for the reviewers’ comments concerning our manuscript entitled"Spatiotemporal differentiation and influencing factors of frost key
date in Harbin municipality in recent 62a" (water-2578506).Those comments are all valuable and very helpful for revising and improving our paper. We have studied comments carefully and have made correction which we hope meet with approval. The main corrections in the paper and the responds to the reviewer's comments are as flowing:
Reviewer 2
â‘ The review's comment: 1. In the introduction section, the progress on frost study in the world should be summarized and highlighted the key finding. The methodology for frost study also should be introduced. Why you conducted this study was also presented clearly.1.
The authors' answer: I've conducted research on the aforementioned topics in the introduction.
â‘¡The review's comment: In the study area data sources and methods section, method should be presented clearly.
The authors' answer: The section on data processing has been revised.
â‘¢The review's comment: In the discussion section, some limitation about this study should be discussed.
The authors' answer: 4.3 Study limitations has now been added
We tried our best to improve the manuscript and made some changes in the manuscript.
We appreciate for Editors/Reviewers’warmwork earnestly, and hope that the correction will meet with approval. Once again thank you very much for your comments and suggestions.
Yours sincerely,
Tiantai Zhang